# Colonic Coffee Phenols Metabolites, Dihydrocaffeic, Dihydroferulic, and Hydroxyhippuric Acids Protect Hepatic Cells from TNF-α-Induced Inflammation and Oxidative Stress

**DOI:** 10.3390/ijms24021440

**Published:** 2023-01-11

**Authors:** Andrea Sánchez-Medina, Mónica Redondo-Puente, Rudolf Dupak, Laura Bravo-Clemente, Luis Goya, Beatriz Sarriá

**Affiliations:** 1Department of Metabolism and Nutrition, Institute of Food Science, Technology and Nutrition (ICTAN-CSIC), Spanish National Research Council (CSIC), José Antonio Nováis 10, 28040 Madrid, Spain; 2Department of Nutrition and Food Science, Faculty of Pharmacy, Universidad Complutense de Madrid (UCM), Plaza de Ramón y Cajal, s/n, 28040 Madrid, Spain; 3Institute of Applied Biology, Faculty of Biotechnology and Food Sciences, Slovak University of Agriculture in Nitra, Trieda Andreja Hlinku 2, 949 76 Nitra, Slovakia

**Keywords:** inflammation, TNF-α, HepG2, metabolites, coffee, cytokines

## Abstract

Coffee presents beneficial health properties, including antiobesity effects. However, its effects on inflammation are controversial. Hydroxycinnamic acids are the main coffee phenolic bioactive compounds. In human bioavailability studies carried out with coffee, among the most abundant compounds found in urine and plasma were the colonic metabolites, dihydrocaffeic (DHCA), dihydroferulic (DHFA), and hydroxyhippuric (HHA) acids. To understand the hepato-protective potential of these three compounds, we tested whether treatment with realistic concentrations (0.5–10 µM) were effective to counteract inflammatory process and oxidative status induced by tumor necrosis factor α (TNF-α). First, we established a novel model of inflammation/oxidation using TNF-α and HepG2 cells. Afterwards, we evaluated the activity of DHCA, DHFA, and HHA against the inflammatory/oxidative challenge through the determination of the inflammatory mediators, interleukins (IL)-6, and IL-8 and chemokines, monocyte chemoattractant protein-1, and macrophage inflammatory protein-1, as well as the levels of biomarkers of oxidative stress, such as reactive oxygen species, reduced glutathione, and the antioxidant enzymes glutathione peroxidase and reductase. Results showed that all three compounds have a potential hepato-protective effect against the induced inflammatory/oxidative insult.

## 1. Introduction

The prevalence of obesity is increasing worldwide and is becoming a major concern for public health. Obesity is complicated with other disorders such as type 2 diabetes mellitus, cardiovascular diseases, metabolic syndrome, non-alcoholic fatty liver disease, hypertension, and arthritis [1,2]. Low-grade inflammation is also associated with obesity and is primarily localized in metabolic tissues, such as white adipose tissue [3,4] and other organs including liver, pancreas, heart, brain, and skeletal muscle [5]. Adipocytokines, such as tumor necrosis factor α (TNF-α) or interleukin (IL)-6, are generally elevated with increasing adiposity and play an important role in the development of metabolic complications in obesity. Adipocytes also express chemokines, such as monocyte chemoattractant protein-1 (MCP-1), macrophage inflammatory protein-1β (MIP-1β), and IL-8. In fact, the circulating MCP-1 and/or IL-8 may be potential candidates linking obesity with obesity-related metabolic complications, such as atherosclerosis and diabetes [6]. The major organ responsible for the glucose production during fasting and metabolic homeostasis in obesity is the liver [7], which has a strong contribution to the development of metabolic inflammation, too [8,9]. The liver may produce pro-inflammatory cytokines that are secreted by resident macrophages, which are associated with impaired hepatic insulin signaling [10] among other effects. Thus, in individuals with obesity, there is a close correlation between adipose tissue inflammation and insulin resistance. However, the mechanisms involved are not completely understood [11,12].

Nowadays, significant attention is given to dietary compounds with anti-inflammatory activity as a natural alternative for the prevention of inflammation-related diseases [13]. Phenolic compounds are naturally occurring and widely distributed in vegetables, fruits, and beverages (tea, coffee, and red wine) that have different biological activities, such as the modulation of enzymatic activity, inhibition of cellular proliferation, or antioxidant and anti-inflammatory properties [14,15,16,17]. Coffee is among the most consumed beverages worldwide and is the major source of phenolic compounds in Western diets [18]. Coffee has shown anti-hyperlipidemic, anti-hyperglicemic, and anti-hypertensive properties [19,20], although the effects on inflammation are inconsistent. According to a recent review [21], coffee consumption may have anti-inflammatory effects, and the discrepancy might be due to large heterogeneity of the available studies, with negative results mostly from short-term, very small trials, with small to moderate consumption of coffee. Based on large studies, it seems that coffee might exert some direct properties related to the reduction of inflammatory biomarkers, as confirmed in most systematic reviews and meta-analyses. In vitro studies in cell culture models suggest the positive, although moderate, effects of coffee could be mediated by hydroxycinnamic acids that have been shown to suppress the expression of interleukins and cell adhesion molecules (vascular, intercellular cell adhesion molecules, and E-selectin) induced by lipopolysaccharide (LPS) [22] or IL-1β, [23] However, the bioavailability of coffee phenols is low, with limited absorption in the small intestine, most of the phenolic metabolites observed in plasma and urine being of colonic origin. In a bioavailability study carried out in humans with a blend of green/roasted coffee (35:65), among the most abundant groups of coffee phenol metabolites of colonic origin (75.7% of total urinary metabolites) were dihydrohydroxycinnamic acids and their phase II derivatives, in addition to feruloylglycine. Among the dihydrohydroxycinnamate metabolites outstand dihydroferulic acid (DHFA) and dihydrocaffeic acid (DHCA) [24,25]. These metabolites reach micromolar concentrations in plasma 5–10 h after the intake of coffee [24,25,26,27]. In addition to these microbial metabolites, derivatives of hydroxyhippuric acid (HHA) were present in both urine and plasma and amounted up to 10 µM in 24 h urine [24,25], although these compounds are not exclusively formed during hydroxycinnamic acid biotransformation.

On the other hand, the absorbed drugs and xenobiotics can cause the over-production of reactive oxygen species (ROS) and free radicals leading the cell to a situation of oxidative stress. This unbalanced condition will mediate cell and tissue damage resulting in inflammatory and fibrotic processes [28]. The liver is particularly susceptible to toxic and oxidative insults since the portal vein brings blood to this organ after intestinal and colonic absorption of nutrients and xenobiotics, including phenolic metabolites. Therefore, studies dealing with the effects of natural dietary antioxidants and their metabolites at a cellular level in cultured hepatic cells are essential. Human HepG2, a well-differentiated transformed cell line, is a reliable model for cultured hepatocyte-type cells used for biochemical, pharmacological, and nutritional studies since they retain hepatocyte morphology and most of its function in culture [29]. The change in the redox potential may indirectly influence transcription and translation through a secondary mechanism, such as cytokine production, and alter the activity of transcription factors [30].

The goal of the present work was to test the potential hepato-protective effects of three of the main colonic metabolites from coffee phenolics: DHFA, DHCA, and HHA. Thus, first, a model of inflammation/oxidative stress-induced damage was developed in liver-derived cells, specifically HepG2. Previous studies have demonstrated that hydroxycinnamic acids [31] are absorbed and metabolized by cultured HepG2 cells, supporting the efficiency of the cell culture line. With the HepG2 cells, we established a novel model of inflammation/oxidation induced by TNF-α. Once the reliability of the model was ensured, the activity of DHFA, DHCA, and HHA against the inflammatory/oxidative challenge was tested through the determination of the inflammatory mediators, cytokines (IL-6 and IL-8), and chemokines (MCP-1 and MIP-1β), as well as the levels of biomarkers of oxidative stress, such as ROS, reduced glutathione (GSH), and the antioxidant enzymes glutathione peroxidase (GPx) and glutathione reductase (GR).

## 2. Results

### 2.1. Establishment of the Inflammation and Oxidation Model in HepG2 Cells

In order to understand the anti-inflammatory effects of DHCA, DHFA, and HHA coffee metabolites, an in vitro model of TNF-α stimulated HepG2 cells with slight modifications of the model previously established by Granado-Serrano et al. [32] was developed. Thus, HepG2 cells were exposed during 6 and 24 h to different concentrations of TNF-α (10, 20, 40, and 60 ng/mL) and the production of pro-inflammatory IL-6 and cell viability were determined. Figure 1 shows that increasing concentrations of TNF-α induced a dose-dependent response of IL-6 up to 40 ng/mL.

However, at 60 ng/mL, the production of IL-6 decreased, and, therefore, this concentration was discarded. The tested doses of 10 to 40 ng/mL TNF-α after 6 h caused no significant changes in cell viability (Figure 2).

In contrast, after 24 h, cell viability decreased with 40 ng/mL treatment. Since the stimulation with 40 ng/mL of TNF-α for 6 h induced a significant increase of IL-6 concentration without decreasing cell viability, this condition was chosen for the following experiments.

To understand the effects on the oxidative status of HepG2 cells in the aforementioned doses (10, 20, 40, and 60 ng/mL) of TNF-α, they were assayed during 2, 4, 6, and 24 h. The highest concentration of TNF-α, 60 ng/mL, evoked a significant ROS over-generation, starting from 2 h and remaining until the last time tested, 24 h. On the contrary, out of the other three tested concentrations, only 40 ng/mL TNF-α induced a significant and reliable increase in ROS at 6 h. Thus, this condition was also the most adequate to establish the model for TNF-α-induced oxidative stress in HepG2 cells (Figure 3).

### 2.2. Direct Effects of DHCA, DHFA, and HHA on HepG2 Cell Viability

Before investigating the anti-inflammatory effects of DHCA, DHFA, and HHA, it was essential to evaluate their cellular safety at the concentrations tested (0.5, 1, 5, and 10 μM). According to crystal violet results, none of the doses of the coffee metabolites induced cytotoxic effects (data shown in Appendix A, Appendix A).

### 2.3. Protective Effects of DHCA, DHFA, and HHA on Inflammation Markers in HepG2 Cells

Once the concentration and time exposure of TNF-α that induced a significant inflammatory/oxidative response was selected, the chemo-protective potential of coffee metabolites was tested in the presence of the TNF-α challenge, but first, the effects of the co-treatment of TNF-α with DHCA, DHFA, and HHA on cell viability were assessed (Figure 4), and no significant cytotoxic effects were observed (*p* > 0–0.05).

Afterwards, the anti-inflammatory potential of the three coffee-derived compounds was analysed through the determination of the inflammatory mediators IL-6, IL-8, MCP-1, and MIP-1β (Figure 5).

It is important to highlight that in our experimental model cells undergo co-treatment with the coffee metabolites and TNF-α, and thus, a direct anti-inflammatory/antioxidant action is expected. Regarding the results of DHCA on inflammation, there were no effects on the production IL-6, MIP-1β, MCP-1, and IL-8. In contrast, the higher concentrations (5 and 10 µM) of DHFA significantly reduced the levels of IL-6, MIP-1β, and IL-8. The third compound tested, HHA, had no significant effects on IL-6 and MCP-1, although the highest concentration provoked a significant decrease of MIP-1β, and, interestingly, HHA counteracted the effects of TNF-α on IL-8 at the four concentrations tested.

### 2.4. Protective Effects of DHCA, DHFA, and HHA on Oxidative Stress Markers in HepG2 Cells

In addition to the anti-inflammatory effects, the levels of biomarkers of oxidative stress, such as ROS and GSH, and the activity of the antioxidant enzymes GPx and GR were measured. The four different concentrations (0.5, 1, 5, 10 µM) of all three metabolites were assayed for ROS and GSH, yet since 0.5 µM had no significant effect in comparison with TNF-α-treated cells, this dose was not assayed for the antioxidant enzyme biomarkers.

Figure 6 shows a dose-dependent decrease in ROS production with 5 and 10 µM DHCA (Figure 6A) and 1–10 µM HHA (Figure 6C) from 4 to 24 h, a result very similar to that of DHFA at 4 and 6 h (Figure 6B).

The results indicate that co-treatment with coffee metabolites dose-dependently prevent the increased generation of ROS induced by TNF-α. Since the more reliable data was obtained at 6 h of co-treatment, this time point was selected for the rest of the oxidative stress markers.

Regarding antioxidant defenses, the concentration of GSH was determined as an index of the intracellular non-enzymatic antioxidant defenses. The treatment of HepG2 cells with 40 ng/mL TNF-α for 6 h induced a mild but significant decrease in the cellular concentration of GSH that was just slightly recuperated by all four doses of DHCA (Figure 7).

However, recovery of TNF-α-decreased GSH levels was observed when cells were co-treated with 5 and 10 µM DHFA and 1–10 µM HHA. These results indicate that all three coffee metabolites, but especially DHFA and HHA, prevent the cell challenging decrease in GSH induced by TNF-α. On the other hand, as major antioxidant enzymes, the evaluation of GPx and GR activity ensures a representative response of the antioxidant system to a stressful challenge. A significant rise in GPx and GR activity was observed in cultured HepG2 cells in response to the treatment with 40 ng/mL TNF-α for 6 h, indicating an active response to face over-generated ROS (Figure 8). The co-treatment with 5 and 10 µM DHFA and HHP metabolites reduced the TNF-α-induced increase of GPx activity, although only the highest concentration of DHCA was able to reduce GPx activity (Figure 8A). Comparably, all three metabolites at the highest doses tested (5 and 10 µM) were able to recover GR activity, reaching values of the control cells (Figure 8B).

## 3. Discussion

Obesity is characterized by a chronic state of low-grade inflammation in which circulating levels of several inflammatory markers, such as IL-6 and MCP-1 [33], and IL-8 [6] are significantly higher compared to the nonobese. Cytokines produced by adipocytes and macrophages, especially TNF-α and IL-6, target the liver. Additionally, the liver maintains a high metabolic rate with an increased production of ROS and other free radicals involved in detoxification reactions, and therefore, it is more susceptible to the potential damaging effects of such compounds and a misbalance of its redox status [34]. However, this organ also receives antioxidants and other bioactive compounds absorbed from the intestinal tract, including colonic metabolites, and therefore, the study of the effect of these bioactive metabolites in cultured liver cells is of great interest. For instance, in previous studies in which oxidative stress was induced with tert-butyl hydroperoxide (t-BOOH) in HepG2 cells, the pre-treatment with the microbial metabolite DHCA (0.2, 1, and 10 µM) prevented cytotoxicity and macromolecular damage induced by t-BOOH, increased the levels of GSH, and recovered ROS and antioxidant enzyme activity. In contrast, DHFA only showed a slight protection against cell cytotoxicity, lipid oxidation, and GSH depletion [35], being the lower activity attributed to the difference in its chemical structure.

TNF-α, in addition to being a major intermediary of the inflammatory process [36], may evoke changes in lipid homeostasis in adipocytes [37] and induce oxidative stress in hepatic cells [32]. To establish the present model of TNF-α-induced inflammation and oxidative stress in HepG2 cells, the results of previous experiments were considered. TNF-α has been used in different cell lines at different concentrations, ranging from 10 ng/mL from 2 to 24 h [38,39,40], up to 40 ng/mL for 24 h [41] or even 50 ng/mL for 18 h, which induced a significant increase of IL-6 in EA.hy926 cells [42]. However, to our knowledge, the TNF-α-induced inflammation and oxidative stress in HepG2 model is novel, as only Granado-Serrano et al. [32] has used TNF-α (6 ng/mL for 30 min) in HepG2 cells to induce inflammation, and they evaluated the effects on the expression of extracellular regulated kinase, c-Jun amino-terminal kinase, production of ROS, and cyclo-oxygenase 2 levels. No other study has used an inflammation/oxidation HepG2 cell model considering the production of inflammatory mediators, such as IL-6, IL-8, MCP-1, and MIP-1β, together with oxidative stress markers (ROS, GSH, GPx, and GR).

Cytokines produced by hepatocytes can regulate hepatocellular functions directly and may influence functions in some liver cells, such as Kupffer cells, which could be an important source of systemically acting pro- and anti-inflammatory molecules [43]. TNF-a is known to induce the expression of several cytokines, namely IL-6. TNF-α and IL-6, by triggering the acute phase response, may lead to the up-regulation of several hepatic inflammatory mediators [44]. In agreement with the present work, the exposure of HepG2 to TNF-α at 40 ng/mL for 6 h stimulated the production of IL-6, MCP-1, MIP-1β, and IL-8 (Figure 5). It is difficult to compare these findings with previous works as in many studies, the induced inflammation in HepG2 cells was analysed through the expression of cytokines [45,46] or, when the production of the cytokine is quantified, it is done mostly using the ELISA technique and not the bead-based multiplex technology used here. These techniques are not comparable because of differences in reagents and recombinant proteins used as standards, and information on assay validation is missing from some commercially available ELISA [47,48]. Another difference that limits comparison is the sample matrix [47] as in other studies primary human and rat hepatocytes [49] or human [20] or rat serum [42] are used. To end, there are studies that use extracts or pure compounds that have not been metabolized even though it is known that the metabolic product should be used to evoke the inflammatory response, as shown in Gomez-Quiroz et al. [50]. These authors showed that acetaldehyde, the metabolic product derived from the oxidative metabolism of ethanol, but not ethanol, stimulates IL-8 production in HepG2 cells. Nevertheless, bearing in mind the distance between the cytokine results presented here and those described in human serum samples after regular consumption of a green/roasted coffee blend [20], there is an agreement as in both studies coffee components did not induce changes on the levels of IL-6 and MCP-1 except for the highest doses of DHFA decreasing IL-6 (Figure 5A). Interestingly, DHFA appeared to be a more biologically active metabolite than DHCA since high doses of DHFA (5 and 10 µM) were able to partially revert the TNF-α-induced increase of IL-6 (Figure 5A), IL-8 (Figure 5D), and MIP-1β (Figure 5B) whilst DHCA showed no effect on any of these inflammatory biomarkers. Similarly, DHFA reduced ROS levels at all the assayed times (Figure 6B), with values even lower than those of the control, unchallenged cells. This antioxidant activity of this metabolite was corroborated assessing its effects recovering the TNF-α-depleted GSH levels (Figure 7) and restoring normal activity of the antioxidant enzymes GPx and GR (Figure 8), again showing a stronger activity compared to DHCA. This is in contrast with the results reported by Baeza et al. [35] in HepG2 cells subjected to an oxidative stress induced by t-BOOH since DHCA was more efficient than DHFA in these conditions.

The evaluation of ROS levels is a reliable index of the oxidative damage to living cells. Slightly increased ROS generation upon treatment with TNF-α denotes a situation of mild oxidative stress that, if prolonged, might induce irreversible cell damage. Reduced ROS generation in cells co-treated with coffee metabolites and TNF-α suggests that these compounds are acting as antioxidants and indicates that enhanced levels of ROS generated during the 6 h of TNF-α-induced stress are being efficiently quenched in co-treated cells, resulting in reduced cell oxidative damage. These findings are in agreement with previous results reporting a protective effect of colonic coffee metabolites from phenolic compounds in endothelial cells [38], as well as cocoa colonic metabolites in the same cell type [51], in pancreatic beta cells [52], renal proximal tubular cells [53], and cardiac cells [54]. Consequently, we can suggest that coffee colonic metabolites DHCA, DHFA, and HHA could restrict ROS production as a primary antioxidant response mechanism to face an oxidative stress.

Another invaluable biomarker of the cell redox status is the concentration of GSH. A balanced GSH concentration is necessary for the cell to face a potential oxidative insult, whereas GSH depletion reveals an increased intracellular oxidation [55]. The decreased GSH concentration induced by TNF-α suggests a situation of potential oxidative stress that might lead to an irreversible oxidative damage in macromolecules (proteins, lipids, and nucleic acids). This dangerous situation was partly or completely prevented by cotreatment with coffee metabolites for 6 h, a response in line with that reported for the flavonol quercetin [32] and green coffee hydroxycinnamic acids [35] in this same cell line, as well as with different colonic metabolites from phenolic compounds in endothelial cells [38]. This result is crucial because maintaining GSH concentration above a critical threshold while facing a stressful situation represents a vital advantage for cell survival.

The activity of antioxidant defense enzymes such as GPx and GR is essential for balancing the intracellular redox status. GPx catalyzes the reduction of cell-damaging peroxide species in coordination with GSH conversion to oxidized glutathione [55], whereas GR recycles oxidized glutathione back to GSH [56], favoring the effective recovery of the steady-state concentration of cellular GSH. The significant increase in the activity of GPx and GR observed after a 6-h treatment with 40 ng/mL TNF-α unmistakably indicates a positive response of the cell defence system to face an oxidative insult [57]. Consequently, at the end of an induced challenge, the antioxidant defense system of cells co-treated with chlorogenic acid metabolites has more promptly returned to a steady-state condition abating cell damage and, thus, allowing the cell to deal with further oxidative insults in better conditions. In the same cell line and experimental conditions comparable to those reported in this study, we and other authors have shown that the flavonoid quercetin [28], the olive oil phenol hydroxytyrosol [58], a coffee melanoidin [59], and the selenium derivative selenomethyl selenium cysteine [60] protect cell damage by preventing the chemically induced over-activity of antioxidant enzymes. Our results contrast with those reported by Gutierrez-Ruiz et al. [30], who described that HepG2 cells treated for 24 h with LPS (1 µg/mL) induced the secretion of IL-1β, IL-6, and IL-8 without changes in GSH or GPx. In fact, in our study, DHFA and HHA, which showed the higher anti-inflammatory effects, particularly at the higher concentrations, also induced a high recovery of GSH. Overall, this data suggests that co-treatment with the colonic metabolites from coffee phenolic chlorogenic acids reduces ROS production enhanced by TNF-α and limits GSH depletion, resulting in a restricted requirement of GPx and GR activity.

Finally, it is worth highlighting the effect of HHA as a biologically active metabolite. To the best of our knowledge, this is the first study on the effects of this colonic metabolite in cultured human hepatic cells. There is only a previous report showing the effect of HHA decreasing TNF-α secretion by LPS-stimulated human peripheral blood mononuclear cells, with no effect on the secretion of IL-6 and IL-1β in this cell model [61]. In the present study, HHA showed to be effective decreasing IL-8 at all tested concentrations (Figure 5D) and MIP-1β at the highest dose (Figure 5B), confirming the anti-inflammatory potential of this colonic metabolite in hepatic HepG2 cells. Similarly, physiologically relevant doses of HHA (1–10 μM) showed antioxidant activity reducing ROS levels (Figure 6C) and recovering endogenous antioxidants (GSH levels, Figure 7, and GPx and GR activity, Figure 8). These results are of special relevance considering that HHA is a colonic metabolite extensively detected in plasma and urine after consumption of different phenolic compounds, not only hydroxycinnamic acids from coffee [24,25] but also flavonoids such as monomeric and oligomeric flavan-3-ols from cocoa, nuts, etc. [61,62], citrus flavanones [63], or berries’ anthocyanins [64] among other fruits. Thus, the anti-inflammatory and antioxidant potential of this common phenolic colonic metabolite is of remarkable interest and merits future research.

The strengths of this work are the use of TNF-α as a natural inducer of inflammation/oxidation in hepatocytes; cell proliferation assays were carried out with untreated cells, cells treated with TNF-α (10, 20, 40, and 60 ng/mL) for 6 and 24 h, and cells treated with phenolic metabolites DHFA, DHCA, and HHA (0.5, 1, 5, and 10 µM) for 6 h. Regarding the co-treatment of TNF-α and the metabolites, the effects on ROS production were measured after 2, 4, 6, and 24 h, as well as the effects on cell viability of the metabolites at 0.5, 1, 5, and 10 μM with TNF-α at 40 ng/mL for 6 h. All these experiments were carried out using serum free medium to avoid interactions with serum proteins. However, a weakness of this work is that it is possible that interactions took place between phenols and TNF-α, which could have affected reactivity. Nevertheless, significant effects were observed in the co-treatment assays.

## 4. Materials and Methods

### 4.1. Chemicals and Reagents

Dulbecco’s modified eagle’s medium (DMEM) and fetal bovine serum (FBS) were from Bio Whitaker Europe (Lonza, Madrid, Spain). DHCA, DHFA, HHA, gentamicin, penicillin, streptomycin, o-phthaldialdehyde (OPT), GR, GSH and oxidised (GSSG) glutathione, nicotine adenine dinucleotide (reduced) (NADH), nicotine adenine dinucleotide phosphate reduced salt (NADPH), 20,70-dichlorofluorescein diacetate (DCFHDA), 2,4-dinitrophenylhydrazone (DNPH), β-mercaptoethanol, guanidine, dithiothreitol (DTT), ethylenediaminetetraacetic acid (EDTA), dimethylsulphoxide (DMSO), t-BOOH, and sodium dodecyl sulphate (SDS) were purchased from Sigma-Aldrich (Madrid, Spain). Bradford reagent was acquired from Bio-Rad (Madrid, Spain). Recombinant murine TNF-α was from PreproTech (Tebu-bio, Madrid, Spain). IL-6 ELISA kit was from R&D Systems (Materlab, Madrid, Spain). The Pro Human Cytokine, Chemokine, and Growth Factor Assay (group I) Multiplex kit was purchased from Bio-Rad (Madrid, Spain). Cell culture dishes were from Falcon (Cajal, Madrid, Spain). Human hepatoma HepG2 cells were a kind gift from Dr. Paloma Martin-Sanz (Instituto de Investigaciones Biomédicas Alberto Sols, CSIC, Madrid, Spain). All other chemicals were of analytical grade.

### 4.2. Cell Culture

Human hepatic HepG2 cells were maintained in a humidified incubator containing 5% CO_2_ and 95% air at 37 °C. They were grown in DMEM F-12 medium and supplemented with 2.5% FBS and 50 mg/L each of gentamicin, penicillin, and streptomycin. ***Establishment of the inflammation/oxidation model in HepG2 cells using TNF-α***

### 4.3. Pro-Inflammatory Treatment

To establish the inflammation model, different concentrations of TNF-α (10, 20, 40 ng/mL), diluted in serum-free culture medium were added to the HepG2 cells during 6 and 24 h. The concentrations and times used were selected according to the literature (see discussion).

### 4.4. Evaluation of Cell Viability

Crystal violet assay was used to determine cell viability. HepG2 cells were seeded at low density (10^5^ cells per well) in 24-well plates. After the TNF-α treatment (10, 20, 40 ng/mL) for 6 and 24 h, cells were incubated with crystal violet (0.2% in ethanol) for 20 min. Plates were rinsed with PBS, and 1% sodium dodecyl sulfate (SDS) was added. The absorbance of each well was measured using a microplate reader at 570 nm (Bio-Tek, Winooski, VT, USA). Results were expressed as percentage of cell viability referred to the absorbance obtained in the control wells (cells treated with serum-free culture medium).

### 4.5. ROS Production

Cellular ROS were quantified by the dichlorofluorescein (DCFH) assay using a microplate reader [28,56]. Cells were seeded in 24-well plates (10^5^ cells per well); DCFH-DA was added for 30 min of incubation, and then wells were washed twice with PBS before the addition of TNF-α (10, 20, 40 ng/mL) for 6 h. Finally, DCFH-derived fluorescence was measured in a fluorescent microplate reader at excitation wavelength of 485 nm and emission wavelength of 530 nm.

### 4.6. Measurement of Interleukin-6 (IL-6) in Cell’s Supernatant

To evaluate the inflammatory effect of TNF-α, cells were seeded in 24-well plates (10^5^ cells per well) in FBS-free medium and after 24 h of incubation, TNF-α (10, 20, 40, 60 ng/mL) was added for 6 h. After that, supernatants were collected for measurement of IL-6 using an immunoenzymatic assay, following the protocol provided by the supplier (Demeditec, Germany). The results were expressed as pg/mL.***Direct effects and protective effects against inflammation/oxidation of DHCA, DHFA and HHA on HepG2 cells***

### 4.7. Direct Effects of DHCA, DHFA, and HHA on HepG2 Cells

HepG2 cells were incubated with DHCA, DHFA, and HHA (0.5, 1, 5, and 10 μM in 0.1% DMSO) for 6 h in serum-free medium and cell viability and ROS production were measured as described in 4.4 and 4.5. DMSO (1%) in deionized water was used to dissolve pure metabolite standards and then diluted with FBS-free DMEM medium (0.1% final DMSO content).

### 4.8. Protective Effects of DHCA, DHFA, and HHA on HepG2 Treated with TNF-α

To study the protective effects of the tested coffee metabolites against inflammation and oxidative stress, HepG2 cells were stimulated with TNF-α (40 ng/mL) together with DHCA, DHFA, and HHA (0.5, 1, 5, and 10 μM in 0.1% DMSO) for 6 h for measuring cell viability and 2 to 24 h for analysing ROS production, as described in 4.4 and 4.5, respectively.

### 4.9. Anti-Inflammatory Effects of DHCA, DHFA, and HHA on HepG2 Cells Treated with TNF-α

To assess the direct anti-inflammatory effect of the phenolic compounds, cells were seeded in 24-well plates (105) in FBS-supplemented medium, and after 24 h of incubation, the cells were treated with different concentrations of DHCA, DHFA, and HHA (0.5, 1, 5, 10 μM) in FBS-free medium with TNF-α (40 ng/mL) for 6 h. Then, cell supernatants were collected and used for determination of IL-6, MCP-1, MIP-1β, and IL-8 using a Bio-Plex kit Pro Human Cytokine, Chemokine, and Growth Factor Assay (group I) (Bio-Rad Laboratories Inc., Hercules, CA, USA), where intra-assay %CV were 7, 9, 8, and 8 and inter-assay %CV were 11, 7, 8, and 8, respectively. All analytes were measured in duplicates on a MAGPIX™ Multiplex reader fitted to a Bio-Plex Pro Wash Station, and software Bio-Plex Manager™ MP (Luminex Corporation, Austin, USA) was used for data processing. Results were expressed as pg/mL supernatant.

### 4.10. Antioxidant Effects Induced by DHCA, DHFA, and HHA in HepG2 Cells Treated with TNF-α

#### 4.10.1. Reduced Glutathione

Cells were plated in 60 mm diameter plates at a concentration of 1.5 × 10^6^ cells/plate. GSH content was evaluated with a fluorometric assay [28,56]. The method takes advantage of the reaction of GSH with OPT at pH 8.0. Fluorescence was measured at excitation and emission wavelengths of 340 nm and 460 nm, respectively. Fluorescence data was interpolated from a standard curve of pure GSH (5–1000 ng).

#### 4.10.2. Antioxidant Enzymes

Cells were seeded in 100 mm diameter plates (2 × 10^6^ cells/plate). Determination of GPx activity was based on the oxidation of GSH by GPx using t-BOOH as a substrate coupled with the disappearance rate of NADPH by GR [28,56]. GR activity was determined by following the decrease in absorbance due to the oxidation of NADPH utilized in the reduction of oxidized glutathione [28,56]. Protein concentration in the samples was measured by the Bradford reagent.

### 4.11. Statistics

Statistical analysis of data obtained from cell culture studies was performed as follows: prior to analysis, the data was tested for homogeneity of variances using the test of Levene; for multiple comparisons, a one-way ANOVA was followed by a Bonferroni test. The level of significance was *p* < 0.05. A SPSS version 24.0 program (SPSS Inc, Chicago, IL) was used.

## 5. Conclusions

Making use of the model of inflammation/oxidation developed here with TNF-α in HepG2 cells, the co-treatment of cells with DHCA, DHFA, and HHA at physiological concentrations (0.5, 1, 5, and 10 µM) showed that DHCA does not have anti-inflammatory effects, as no changes were observed on IL-6, MIP-1β, MCP-1, and IL-8, in contrast to DHFA and HHA, which, at higher concentrations, showed remarkable anti-inflammatory potential, also shown by HHA decreasing IL-8 from the lower concentrations studied. Regarding antioxidant effects, the three compounds tested reduced ROS production enhanced by TNF-α and limited GSH depletion, particularly higher concentrations of DHFA and HHA, resulting in a dose dependent restricted requirement of GPx and GR activity that was similar with DHCA, DHFA, and HHA. In conclusion, the three major colonic metabolites of coffee phenolic compounds showed a remarkable potential hepato-protective effect against the TNF- α -induced inflammatory and oxidative conditions. This is the first study reporting the anti-inflammatory and antioxidant properties of the widespread colonic metabolite HHA in cultured hepatic cells.

## Figures and Tables

**Figure 1 ijms-24-01440-f001:**
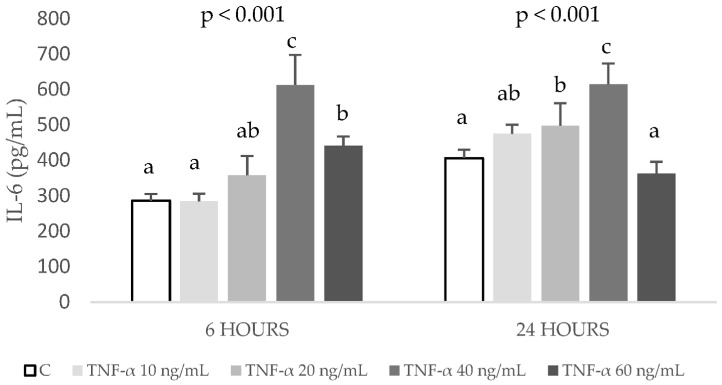
Production of interleukin-6 (IL-6) after exposure to 0 (control, C), 10, 20, 40 and 60 ng/mL of TNF-α during 6 and 24 h in HepG2 cell cultures. Results, expressed as pg/mL, are means ± SD (n= 3–6). The *p* values above the figures represent ANOVA results and the different letters denote statistical differences within the same time according to the Bonferroni test (*p* < 0.05).

**Figure 2 ijms-24-01440-f002:**
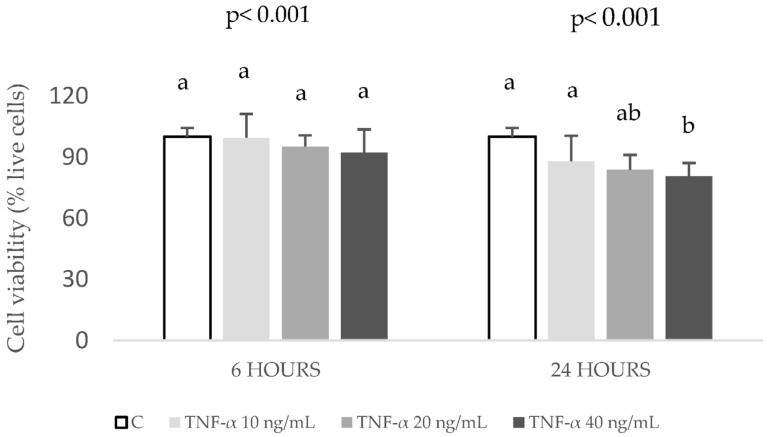
Effects of TNF-α at 0 (control, C), 10, 20 and 40 ng/mL, during 6 and 24 h, on cell viability in HepG2 cultures. Results, expressed as % of live cells, are means ± SD (n = 3–6). The *p* values above the figures represent ANOVA results and the different letters denote statistical differences within the same time according to the Bonferroni test (*p* < 0.05).

**Figure 3 ijms-24-01440-f003:**
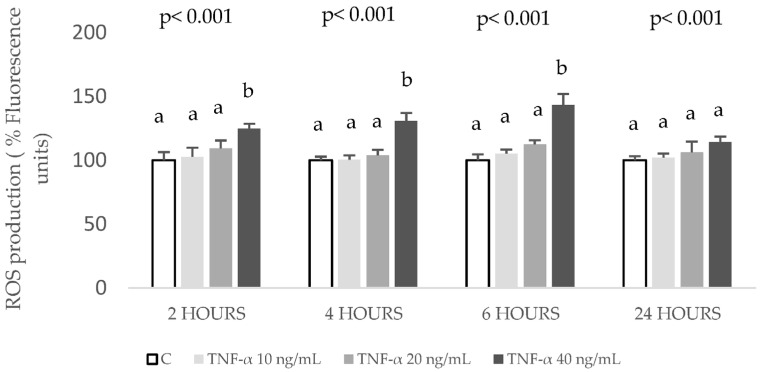
Effect of (TNF-α) on reactive oxygen species (ROS) production in HepG2 cells. Concentrations of 0 (control, C), 10, 20, and 40 µM of TNF-α were added to HepG2 cultures and ROS levels were evaluated at the noted time points. Data are means ± SD of 4 different samples per condition. The *p* values above the figures represent ANOVA results and the different letters denote statistical differences according to the Bonferroni test within the same time point (*p* < 0.05).

**Figure 4 ijms-24-01440-f004:**
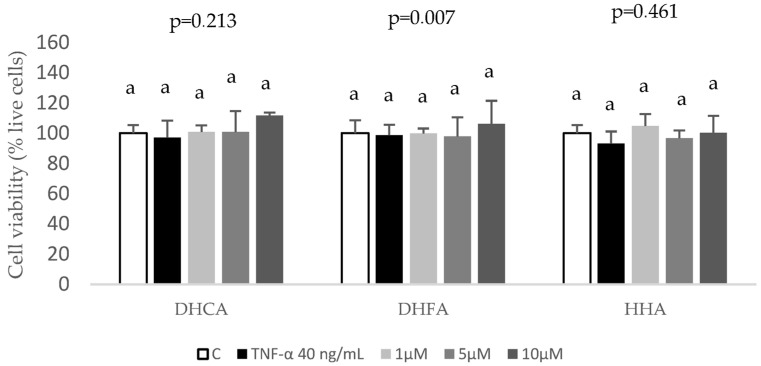
Effect of coffee microbial metabolites dihydrocaffeic acid (DHCA), dihydroferulic acid (DHFA), and hydroxyhippuric acid (HHA) at 0 (control, C), 1, 5 and 10 μM with cotreatment of TNF-α at 40 ng/mL during 6 h on cell viability in HepG2 cultures. Results, expressed as % of live cells, are means ± SD (n = 4). Within the same compound, there were no significant differences according to the ANOVA and the Bonferroni test (*p* < 0.05).

**Figure 5 ijms-24-01440-f005:**
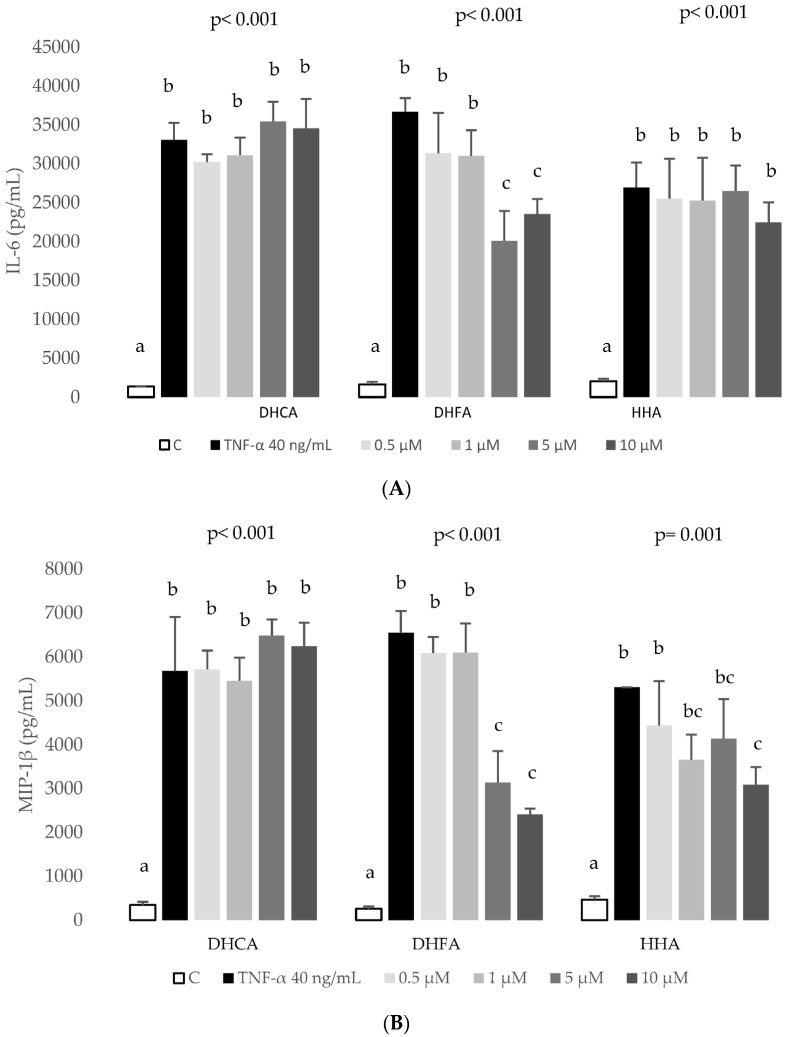
Effect of coffee microbial metabolites DHCA, DHFA, and HHA at 0 (C), 0.5, 1, 5 and 10 μM with cotreatment of TNF-α at 40 ng/mL during 6 h on the production of interleukin-6 (IL-6; (**A**)), macrophage inflammatory protein 1 (MIP-1β; (**B**)), monocyte chemoattractant protein 1 (MCP-1; (**C**)) and interleukin-8 (IL-8; (**D**)) in HepG2 cells. Results, expressed as pg/mL, are means ± SD (n = 4). The *p* values above the figures represent ANOVA results and different letters denote statistical differences according to the Bonferroni test within the same compound (*p* < 0.05).

**Figure 6 ijms-24-01440-f006:**
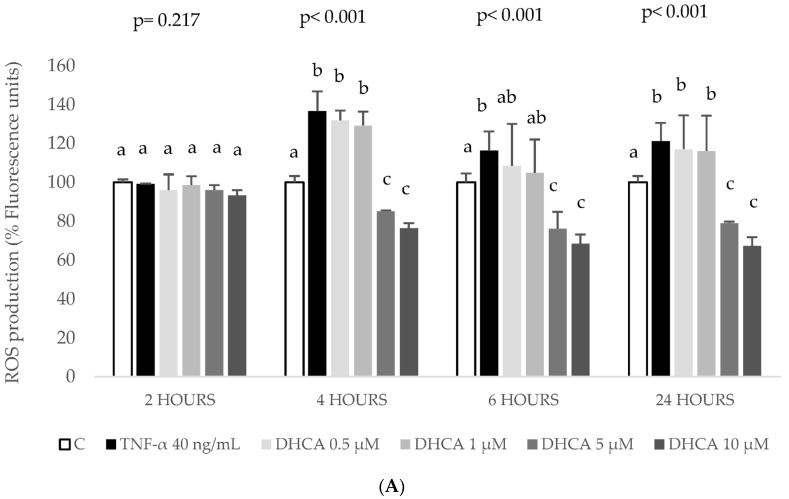
Effect of coffee microbial metabolites dihydrocaffeic acid (DHCA, (**A**)), dihydroferulic acid (DHFA, (**B**)), and hydroxyhippuric acid (HHA, (**C**)) at 0 (control, C), 0.5, 1, 5 and 10 μM on reactive oxygen species (ROS) over-production induced by tumor necrosis factor-α (TNF-α). HepG2 cells were co-treated with noted concentrations of metabolites and TNF-α (40 ng/mL during 6 h). ROS concentration was determined at the referred time points. Data are means ± SD of 4 different samples per condition. The *p* values above the figures represent ANOVA results and the different letters denote statistical differences according to the Bonferroni test within the same time point (*p* < 0.05).

**Figure 7 ijms-24-01440-f007:**
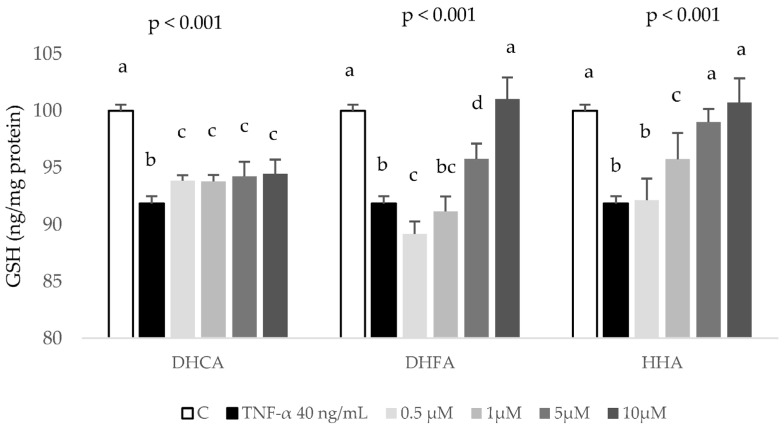
Effect of coffee microbial metabolites DHCA, DHFA, and HHA at 0 (control, C), 0.5, 1, 5 and 10 μM on depleted reduced glutathione (GSH) levels induced by TNF-α. HepG2 cells were cotreated with noted concentrations of the metabolites and TNF-α (40 ng/mL for 6 h) and GSH concentration was determined. Data are means ± SD of 3–4 different samples per condition. The *p* values above the figures represent ANOVA results and the different letters denote statistical differences according to the Bonferroni test within the same compound (*p* < 0.05).

**Figure 8 ijms-24-01440-f008:**
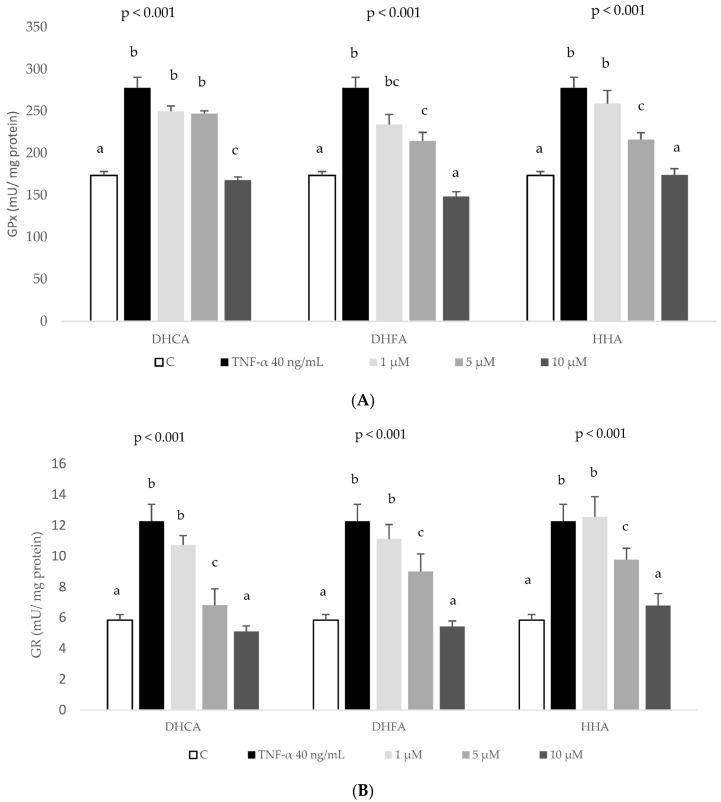
Effect of coffee microbial metabolites DHCA, DHFA, and HHA at 0 (control, C), 1, 5 and 10 μM on enhanced antioxidant enzymes activity induced by TNF-α. HepG2 cells were cotreated with noted concentrations of metabolites and TNF-α (40 ng/mL for 6 h) and glutathione peroxidase (GPx; (**A**)) and glutathione reductase (GR; (**B**)) activity were assayed. Data are means ± SD of 3–4 different samples per condition. The *p* values above the figures represent ANOVA results and the different letters denote statistical differences according to the Bonferroni test within the same compound (*p* < 0.05).

## Data Availability

The data presented in this study are available on request from the corresponding author.

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
