# Peer review of "Colonic Coffee Phenols Metabolites, Dihydrocaffeic, Dihydroferulic, and Hydroxyhippuric Acids Protect Hepatic Cells from TNF-α-Induced Inflammation and Oxidative Stress"

_ijms, 2023, doi:10.3390/ijms24021440_

Round 1
Reviewer 1 Report
The goal of the present work was to test the potential hepato-protective effects of three of the main colonic metabolites from coffee phenolics: DHFA, DHCA, and HHA. With the HepG2 cells, the authors established a novel model of inflammation/oxidation induced by TNF-α. The activity of DHFA, DHCA, and HHA against the inflammatory/oxidative challenge in hepatocytes was tested through the determination of the inflammatory mediators, cytokines (IL-6 and IL-8) and chemokines MCP-1 and MIP-1β, as well as the levels of biomarkers of oxidative stress, such as ROS, reduced glutathione (GSH), and the antioxidant enzymes glutathione peroxidase (GPx) and glutathione reductase (GR).
The strengths and this manuscript is the use of TNF-α as an natural inducer of inflammation/oxidation in hepatocytes. The weaknesses is that the authors didn’t consider the posible interaction between polyphenols and TNF-α which could affect reactivity of it.
Polyphenols are compounds which could interact with proteins and change them activity. TNF-α is a protein which could be affected by polyphenols. Ones both compounds are mixed in medium this could affect them reactivity. In order to prevent and search polyphenols reactivity they should be mixed with the medium separately before and after the addition of the TNF-α 0, 15, 30, 60 minutes, at least. This will give the authors important results about a possible effect/or no effect between both compounds.
Author Response
Thank you for your comments. In agreement with your observations, the following paragraph has been included in the manuscript.
Strengths of this work are the use of TNF-α as a natural inducer of inflammation/oxidation in hepatocytes; cell proliferation assays were carried out with untreated cells, cells treated with TNF-α (10, 20, 40 and 60 ng/mL) for 6 and 24 hours and cells treated with DHFA, DHCA and HHA (0.5, 1, 5 and 10 µM) for 6 hours. Regarding the co-treatment of TNF-α and the metabolites, the effects on ROS production were measured after 2, 4, 6 and 24 hours, as well as the effects on cell viability of the metabolites at 0.5, 1, 5 and 10 μM with TNF-α at 40 ng/mL for 6 hours. All these experiments were carried out using serum free medium to avoid interactions with serum proteins. However, a weakness of this work is that it is possible that interactions took place between phenols and TNF-α, which could have affected reactivity. Nevertheless, significant effects were observed in the co-treatment assays.
Reviewer 2 Report
The manuscript deals with the evaluation of the protective effect of colonic metabolite of coffee hydroxycinnamic acids on the inflammatory process and oxidative status induced by TNF-α in hepatic cells. The topic is of high importance and practical interest.
Dihydrocaffeic, dihydroferulic, and hydroxyhippuric acids have been studied. The effectivity of their action has been considered using a wide range of parameters allowed to make reliable conclusions. The inflammatory mediators (IL-6 and IL-8, MCP-1, and MIP-1β) and biomarkers of oxidative stress (ROS, GSH, GPx, and GR) have been considered.
The manuscript is well-written. The experiment is logically designed and presented. The data obtained are well-discussed and compared to reported data. Conclusions are supported with results obtained.
The manuscript can be accepted to publication after minor revision.
1. Abbreviations need careful revision. Many of them are inserted to the text several times. Both abbreviated and full names are existed. Abbreviation used 1-2 times throughout the text should be removed.
2. Section 2.2:
a) text on line 153 is detached from line 152;
b) line 156, data can be presented in the Supporting material file
3. Figure 4 caption needs revision. There are no data for the concentration of 0.5 μM in the figure. Please, remove this information.
4. The resolution of the figures is insufficient.
Author Response
Thank you for your comments which have improved the quality of our work. Track-changes have been used to identify the modifications we have made following your suggestions.
- Abbreviations need careful revision. Many of them are inserted to the text several times. Both abbreviated and full names are existed. Abbreviation used 1-2 times throughout the text should be removed.
Abbreviations have been carefully revised, so that they are inserted only once.
- Section 2.2:
- a) text on line 153 is detached from line 152;
Lines 152 and 153 are attached.
- b) line 156, data can be presented in the Supporting material file
The data is presented in a supporting material file
- Figure 4 caption needs revision. There are no data for the concentration of 0.5 μM in the figure. Please, remove this information.
The concentration of 0.5 μM in the caption of figure 4 has been removed.
- The resolution of the figures is insufficient
The resolution of the figures has been improved.
